# Recombinant Chromosome 7 Driven by Maternal Chromosome 7 Pericentric Inversion in a Girl with Features of Silver-Russell Syndrome

**DOI:** 10.3390/ijms21228487

**Published:** 2020-11-11

**Authors:** Ilaria Catusi, Maria Teresa Bonati, Ester Mainini, Silvia Russo, Eleonora Orlandini, Lidia Larizza, Maria Paola Recalcati

**Affiliations:** 1Laboratorio di Citogenetica e Genetica Molecolare, Istituto Auxologico Italiano, IRCCS, 20149 Milano, Italy; i.catusi@auxologico.it (I.C.); estermainini@yahoo.it (E.M.); s.russo@auxologico.it (S.R.); l.larizza@auxologico.it (L.L.); 2Ambulatorio di Genetica Medica, Istituto Auxologico Italiano, IRCCS, 20149 Milano, Italy; mtbonati1@gmail.com (M.T.B.); eleonora.orlandini2@studio.unibo.it (E.O.)

**Keywords:** Silver-Russell syndrome, *GRB10* gene, array-CGH, chromosome 7p duplication, chromosome 7 pericentric inversion, recombinant chromosome, *AUTS2* gene

## Abstract

Maternal uniparental disomy of chromosome 7 is present in 5–10% of patients with Silver-Russell syndrome (SRS), and duplication of 7p including *GRB10* (Growth Factor Receptor-Bound Protein 10), an imprinted gene that affects pre-and postnatal growth retardation, has been associated with the SRS phenotype. Here, we report on a 17 year old girl referred to array-CGH analysis for short stature, psychomotor delay, and relative macrocephaly. Array-CGH analysis showed two copy number variants (CNVs): a ~12.7 Mb gain in 7p13-p11.2, involving *GRB10* and an ~9 Mb loss in 7q11.21-q11.23. FISH experiments performed on the proband’s mother showed a chromosome 7 pericentric inversion that might have mediated the complex rearrangement harbored by the daughter. Indeed, we found that segmental duplications, of which chromosome 7 is highly enriched, mapped at the breakpoints of both the mother’s inversion and the daughter’s CNVs. We postulate that pairing of highly homologous sequences might have perturbed the correct meiotic chromosome segregation, leading to unbalanced outcomes and acting as the putative meiotic mechanism that was causative of the proband’s rearrangement. Comparison of the girl’s phenotype to those of patients with similar CNVs supports the presence of 7p in a locus associated with features of SRS syndrome.

## 1. Introduction

Silver-Russell syndrome (SRS, #180860) is associated with all the following six clinical criteria, of which four must be present for a diagnosis to be made: pre- and postnatal growth retardation, relative macrocephaly, prominent forehead, body asymmetry, and feeding difficulties (Netchine-Harbison clinical scoring system, NH-CSS) [1]. The most common underlying mechanisms are loss of methylation on chromosome 11p15 (30–60% of patients) and chromosome 7 uniparental disomy (5–10% of patients). *GRB10* (Growth Factor Receptor-Bound Protein 10), an imprinted gene whose effects include pre- and postnatal growth retardation, is located at 7p12. Duplications involving *GRB10* on the maternal allele are described in patients with growth delay and SRS features [2,3,4], whereas those on the paternal allele are associated with overgrowth [5]. To date, only five patients with maternal 7p duplication including *GRB10* have been described in the literature and some of them are related [2,3,4].

Here, we report on a 17-year-old girl referred to our lab with some features of SRS syndrome associated with severe intellectual disability (ID). Array-CGH (Array based Comparative Genomic Hybridization) analysis detected a complex rearrangement composed of an ~12.7 Mb gain at 7p13-p11.2 and an ~9 Mb loss at 7q11.21-q11.23. By FISH (Fluorescence In Situ Hybridization) analysis, the proband’s mother was found to carry an apparently balanced, pericentric inversion that could have triggered the complex rearrangement.

## 2. Results

### 2.1. Clinical Report

The girl was born to healthy unrelated parents after 40 weeks of gestation. Labor was induced because of polyhydramnios. The girl’s birthweight was 2.970 g (18th centile), her birth length was 49 cm (29th centile), and her occipitofrontal circumference (OCF) was 35 cm (77th centile). Apgar score was 9 and 10 at 1 and 5 min, respectively.

As for developmental milestones, at 6 months she was recognized to have poor smiling and reduced interest in parents and response when called by her name. Her movements tended to be stereotyped. She was therefore started on psychomotor education. She sat and walked unaided at 11 and after 24 months, respectively. At 17 months she said her first words. At 24 months, she could repeat every word. We were not able to further detail speech and language development. At the age of 4–5 years, a stereotyped movement with her head—shaking it as if to say ‘no’—appeared and was almost always present at the last clinical genetic evaluation at 17 years and 4 months.

The first cognitive evaluation, carried out at 4 years and 4 months, showed a full scale IQ (FSIQ) of 77 (Griffiths), corresponding to a slight intellectual disability (ID). However, at a recent neuropsychological assessment, her FSIQ was <40 (WISC, Wechsler Intelligence Scale for Children), suggestive of a severe ID, with strengths in language (VCI, verbal comprehension index = 54), as well as in computation and reading, and weaknesses in visual-perceptive reasoning as well as in working memory (PRI, perceptual reasoning index = 41).

Her growth (length and weight) began to slow down at 3 months of age, and her height curve progressively moved down to approximately −3 SDS at 6 months of age. She always had a poor appetite. At the age of 4 years she still ate pureed food.

Due to a growth hormone (GH) deficiency (GH = 5 ng/mL at GHRH-arginine test), she was started on rhGH therapy at a dosage of 0.6 mg/day (6 days a week) from the age of 11 years and 4 months. At that age, her diet was still selective and she suffered from constipation. Under hormone therapy, her stature varied between −3.63 and −2.80 SDS, with a growth rate fluctuating between 4.4 and 10.2 cm per year. Menarche occurred at 12 years and 9 months. Her bone age (BA) was always delayed compared to her chronological age (CA) until she was 10 years old, when the BA vs. CA distance began to decrease. GH was stopped at 15 years and 2 months; when her height was 144 cm (−2.81 SDS). Hyperinsulinism was diagnosed the year before.

In addition, she had flat foot arches for which she underwent surgery on the left foot at age 15, a year after the discovery of scoliosis.

### 2.2. Cytogenetics Analysis

The array-CGH analysis showed two copy number variants (CNVs) on chromosome 7: an ~12.7 Mb gain at 7p13-p11.2 (arr[GRCh37] 7p13p11.2(44114508_56786860)) and an ~9 Mb loss at 7q11.21-q11.23 (arr[GRCh37] 7q11.21q11.23(63374309_72365957)). The ~5.6 Mb pericentromeric segment located between the two CNVs (nucleotides 57262017 to 62833642) had a normal copy number (Figure 1a). Subsequent FISH experiments with the BAC (Bacterial Artificial Chromosome) probes RP11-48B18 (7p12.3) and RP11-80L24 (7p12.2) showed that the duplicated region mapping on the long arm of the rearranged chromosome 7 was in an inverted orientation (Figure 1b). Moreover, BAC probes RP11-114G11 (7p11.2) and RP11-45N18 (7q11.21) established that the regularly represented pericentromeric region of 5.6 Mb was inverted too (Figure 1c). Conventional chromosome analyses of the parents were normal but the FISH analysis of RP11-48B18 and RP11-90C3 (7q11.23) BAC probes on the mother’s metaphases showed a pericentric inversion (Figure 1d). The same FISH pattern was observed in the proband’s sister. In order to achieve a better definition of the inversion breakpoints, a panel of BAC FISH probes was used. The breakpoint on the q arm localized at 7q11.23 between nucleotides 72320083 and 72757459. Unfortunately, the inversion breakpoint on the short arm could be only located distally to the BAC probe RP11-147I22 (7p13) (data not shown).

FISH patterns on father’s and maternal grandmother’s metaphases were normal (not shown). A sample from the maternal grandfather was not available.

### 2.3. Molecular Genetic Analyses

The analysis of the segregation pattern of Single Nucleotide Polymorphisms (SNPs) from the parents to the proband and her sister demonstrated that the deletion occurred on maternal chromosome 7 (Table 1). Moreover, the familial segregation of polymorphic microsatellite markers mapped within the 7p13-p11.2 chromosomal region allowed us to assess the presence of maternal heterodisomy in the proband, because both mother alleles were present for this region (Table 2).

To investigate whether the duplication impaired the methylation pattern at the GRB10:alt-TSS-DMR, we performed MS-MLPA analysis which revealed hypermethylation of the maternal allele in the proband and a normal pattern in her sister carrying the pericentric inversion. The CNV analysis obtained with the assay confirmed the duplication of the 7p12.2 cytoband in the proband (data not shown).

## 3. Discussion

### 3.1. Genotype–Phenotype Correlation

There are a few comparable patients with maternal 7p13-p11.2 duplication in the literature; their clinical and genomic data are reported in Table 3 and Figure 2. The survey of reported duplications patients was guided by the physical map of the region kindly provided by Dr. David Monk (Bellvitge Biomedical Research Institute, Barcelona, Spain).

We reviewed the clinical reports of all patients on the basis of the Netchine Harbison clinical scoring systems (NH-CSS) [1]. None of the patients satisfied the clinical diagnosis of SRS, which requires the presence of four criteria [1], as they exhibited only one or two NH-CSS items.

The most distinctive diagnostic criterium for SRS, relative macrocephaly at birth (criterium n.3, Table 3), was present in our patient, absent in LB, and not available in the remaining patients, whereas a protruding forehead (n.4) was exhibited only by one patient (DP) out of six. One out of four patients was small for gestational age (n.1), and three out of four showed postnatal growth failure (criterium n.2). It is noteworthy that none exhibited body asymmetry. Therefore, what seems to distinguish this series of patients is postnatal growth failure (criterium n.2), which was accurately documented in the clinical report of the patient described here, as well as some additional SRS features, such as fifth finger clinodactyly, micrognatia, hypoglycaemia, a triangular face, a down-turned mouth, and excessive sweating (Table 3). The final height ranged from −2 SDS (TB, HC) to −2.81 SDS (present patient). The lowest height during growth was −3.56 SDS (LB at 6 years and 3 months).

Data from our patient support the link between *GRB10* maternal duplications and growth delay. Since other genes involved in human growth are included in the duplicated 7p12.3 region (*IGFBP1*, *IGFBP3*, and *EGFR*), their influence on the growth phenotype cannot be excluded. To our knowledge, no isolated duplications of these genes have been described either in the literature or in the DECIPHER database [6]. Moreover, none of these genes have actually been reported in the OMIM database as being related to growth defects [7].

The role of *GRB10* in SRS is still controversial, because of its peculiar expression and imprinting profile [8]. Both mice and humans display a conserved CpG island with a DMR methylated in the maternal germline, but these data do not match the expression profile. While mouse *Grb10* shows maternal-specific expression, the human *GRB10* gene codes for various transcripts that are biallelically expressed almost in all tissues. Paternal monoallelic expression of a specific isoform in the fetal brain and maternal monoallelic expression of a specific isoform in fetal muscle have been reported [9]. The overall phenotype of our case is consistent with the idea that an imbalance of fetal *GRB10* expression in the brain and muscle might contribute to mental impairment and growth disorder.

With the exception of AC and his mother, all 7p13-p11.2 duplicated patients exhibited a neurodevelopmental disorder (NDD), supporting the presence of development-related genes in the region [2,3,4]. Two genes leading to impaired cognition when mutated map in the common duplicated region: *CAMK2B*, which is associated with a dominant form of ID (MRD54, # 617799), and *DDC*, which is involved in Aromatic l-amino acid decarboxylase deficiency (# 608643) [7]. No evidence of a dosage effect of these genes due to copy number gain is indicated by the literature. Further studies need to be performed to clarify which genes in the duplicated region are linked to NDD. Normotypical development of AC and his mother was inferred by the absence of relative details in the clinical report and, for AC, by the effect of mosaic duplication.

Regarding our patient, the most relevant rearrangement for ID seems to be the 7q11.21-q11.23 deletion, as this region encompasses 25 genes including *AUTS2* [10]. The severity of ID due to *AUTS2* haploinsufficiency usually varies from moderate to borderline according to the involvement of the C terminal portion of the protein, as the gene has different isoforms. ID in the patient was classified as severe, as expected by the complete *AUTS2* deletion as well as by the cumulative effect of *AUTS2* deletion and duplication of the above mentioned genes involved in cognition (*CAMK2B* and *DDC)*. Accordingly, the patient’s developmental milestones fit those reported in patients carrying the deletions of the whole gene. Moreover, *AUTS2* is associated with susceptibility to autism traits, including stereotypic movements and hypersocial behavior. Our patient was described to exhibit autism traits in infancy; however, she has never been systematically assessed for autism spectrum disorders. Feeding difficulties may occur in AUTS2 syndrome; therefore, our patient’s clinical signs may have resulted from the co-contribution of *AUTS2* deletion and *GRB10* hypermethylation. Conversely, she did not exhibit microcephaly, a feature found in 60% of patients affected by AUTS2 syndrome, as the patient’s OCF at 14 years (cm 49.5) was in the percentile curve expected for her stature.

The patients’ CNVs and gene localization are shown in the map in Figure 2.

### 3.2. Rearrangement Mechanism

As mentioned, the proband’s mother and sister are carriers of a pericentric chromosome 7 inversion. Based on the FISH results, we propose that the inversion breakpoints were localized at the distal breakpoint of the daughter’s gain (7p13) and at the distal breakpoint of the daughter’s deletion (7q11.23). The inversion had an estimated extent of about 28.6 Mb (sized about 18% of the entire chromosome 7). As the inversion encompasses the centromere and the pericentromeric regions, despite its length, it was not cytogenetically visible. The inversion, not inherited by the maternal grandmother (data from the grandfather were unfortunately not available), has not been described as polymorphic in humans [11,12].

Chromosome 7 is highly enriched in segmental duplications (or Low Copy Repeats, LCRs), some of which, with a high degree of reciprocal homology, were found to be located at the breakpoints of both the mother’s inversion (LCR-A and B) and the daughter’s CNVs (LCR-C and D) (Figure 2). LCR-A and LCR-B may have therefore mediated the mother’s inversion.

We hypothesize that during gametogenesis pairing of the paralogous sequences (LCR-C and LCR-D) in the mother, mapping to the inversion endpoints might have perturbed correct meiotic chromosome segregation, leading to an unbalanced outcome, according to the model proposed in Figure 3. During gametogenesis in the mother, mispairing of LCR-C on the normal chromosome 7 and LCR-D on the inverted chromosome 7, quite symmetrically with respect to the centromere, might have triggered a non-allelic homologous recombination (NAHR) event (red cross), leading to the complex rearrangement described in the present work characterized by (1) an inverted duplication between LCR-A and LCR-C/D (blue); (2) an inverted normal copy number pericentromeric region between the recombinant LCR-C/D and LCR-C (white); and (3) a deletion of the segment between LCR-B and D (red in the maternal chromosome).

Sperm studies are useful tools to evaluate the segregation products of male carriers of a chromosomal aberration. In cases of carriers of a pericentric inversion, studies have suggested that no recombinant gametes are expected in carriers of inversions <50 Mb that involve a segment shorter than 30–40% of the whole chromosome length [13,14,15,16]. Indeed, short inversions usually tend to form a “balloon” at meiosis and do not align homologous regions with few possibilities of NAHR events. In sperm studies, the estimation of the risk of transmitting a recombinant chromosome is based on the observation of unbalanced male gametes: however, one should keep in mind that (1) male and female carriers could have different reproductive risks and (2) inversion on different chromosomes could behave in different ways at meiosis. A chromosome specific correlation between the frequency of recombinants and inversion size was performed by Luo Y. et al. (2014) for chromosome 1 [16]. The study suggested that no recombinants are expected for inversion lengths < 18%. The only sperm study conducted for a chromosome 7 pericentric inversion carrier was carried out by Navarro et al. (1993) [17]. In the reported case, the inverted segment was 118 Mb (75% of the chromosome length) and 25% of recombinants were observed. However, a single study is not sufficient to establish a correlation. As suggested by the present work, carriers of inversions involving LCR-rich chromosomes might have a reproductive risk higher than that of carriers of inversions of other chromosomes due to possible NAHR events within the inverted regions. Further chromosome-specific correlation studies are needed to help counselors to offer informed genetic counselling in such cases to couples who might want to consider prenatal or pre-implantation diagnosis.

## 4. Materials and Methods

This work was included in a research project approved by the ethics committee of Istituto Auxologico Italiano on 12 April 2017 (RC 08C723).

Cytogenetic analysis was performed on peripheral blood lymphocytes using QFQ banding techniques on metaphase chromosomes obtained by standard procedures. The BAC probes used in the FISH study were selected according to the University of California, Santa Cruz (UCSC) Genome Browser (GRCh37/hg19, release February 2009). BAC probes underwent nick translation labelled with biotin-dUTP (Deoxyuridine triphosphate) (Merck KGaA, Darmstad, Germany) or Cy3-dUTP (Merck KGaA, Darmstad, Germany), and detection of the biotin-dUTP labelled probe was done with DEAC-streptavidin (7-Diethylaminocoumarin-3-carboxylic acid). The panel of BAC FISH probes used was CTD-2256C5, PR11-146I22, CTD-2591O16, CTD-2069P19, RP11-48B18, RP11-80L24, RP11-114G11, RP11-45N18, RP11-90C3. Genomic DNA was purified for molecular studies from peripheral blood mononucleate cells by using the GenElute™Blood Genomic DNA kit (Merck KGaA, Darmstad, Germany) according to the supplier’s instructions. Array-CGH analysis was conducted by using the SurePrint G3 Human CGH Microarray kit 8x60K with a mean resolution of ~130 kb (Agilent, Santa Clara, CA, USA). SNP segregation analysis was conducted using the SurePrint G3 Human CGH+SNP Microarray kit 4x180K (Agilent, Santa Clara, CA, USA). Protocols provided by Agilent were followed with no modifications. The arrays were analyzed with Agilent Cytogenomics software (v 3.0.6.6). Nucleotide designations were made according to the GRCh37/hg19 build of the human genome.

Segregation analysis of polymorphic loci was performed on the patient, her parents, her sister, her maternal grandmother, and her maternal uncle using microsatellites mostly spanning the chromosome 7 duplicated region. The following polymorphic markers were used: D7S519 (7p13), D7S2422 (7p12.1), D7S2467 (7p12.1), D7S506 (7p12.1), and D7S2552 (7p11.2).

Primer sequences and positions are available on UCSC Genome Browser (release March 2006; http://genome.ucsc.edu/cgi-bin/hgGateway).

Polymerase chain reaction (PCR) assays were developed and optimized to amplify each fragment containing a marker of interest in a 15 μL volume using TaqGold 0.15 u (Biorad, Hercules, CA, USA) and GeneAmp PCR Gold Buffer 10× (Biorad, Hercules, CA, USA) with 50–100 ng of DNA and 0.1 μM of each primer. Final products were prepared for injection on the ABI PRISM 310 Genetic Analyzer (Applied Biosystems, Foster City, CA, USA) using deionized formamide, 0.5 μL of GS ROX500 size standard (Applied Biosystems, Foster City, CA, USA) and 1.0 μL of PCR products. Products were separated by POP7 and analyzed by GeneScan.

MLPA analysis was performed with MCR-HOLLAND (Amsterdam, The Netherlands) SALSA MS-MLPA Probemix ME032 UPD7-UPD14 kit.

As the PAC clones employed to characterize the chromosome rearrangements of the literature patients were no longer available in UCSC, our cross-comparison was guided by a physical map of the region kindly provided by Dr David Monk (Bellvitge Biomedical Research Institute, Barcelona, Spain). Accordingly, we located the approximated proximal breakpoint of patient DP (Monk et al., 2000; 2002) at UCSC D7S668, that of patients TB/LB (Joyce et al., 1999; Monk et al., 2002) at D7S2422, and that of HC/AC (Monk et al., 2002) at the *RALA* gene. We mapped the distal breakpoint of patient DP at D7S22, that of patients TB/LB at *GLI3*, and that of patients HC/AC at *PHKG1*.

## 5. Conclusions

Integrated genomic targeted molecular cytogenetics and genetic analyses conducted on a girl and her family permitted us to (1) provide evidence for the link between *GRB10* maternal duplication and SRS clinical features and (2) provide data to allow us to reflect on the reproductive risk of carriers of relatively small pericentric inversions. The complex case here solved highlights the relevance of genetic counseling in assessing not only the size of the chromosomal rearrangement (pericentric inversion) but also the genomic architecture of the involved chromosomal region.

## Figures and Tables

**Figure 1 ijms-21-08487-f001:**
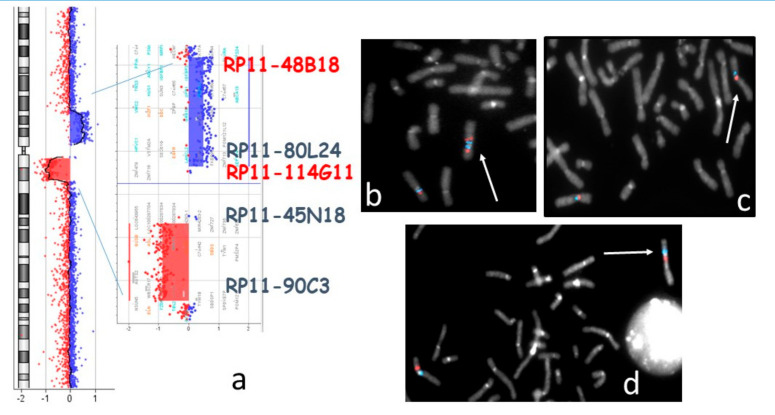
(**a**) Proband’s array-CGH profile of chromosome 7 with enlargement of the regions involved in Copy Number Variants (CNVs). The localization of the probes used for FISH experiments is indicated. (**b**,**c**) Bacterial Artificial Chromosome (BAC) FISH analysis of the proband’s metaphases: (**b**) RP11-48B18 (red), RP11-80L24 (blue). The white arrow points to the duplicated signals and inverted orientation on 7q of the two 7p probes: (**c**) RP11-114G11 (red), RP11-45N18 (blue). The white arrow indicates the inverted orientation of the pericentromeric region on the rearranged chromosome 7; (**d**) BAC FISH on the proband’s mother’s metaphases: 48B18 (red), RP11-90C3 (blue). The white arrow points to chromosome 7 with pericentric inversion.

**Figure 2 ijms-21-08487-f002:**
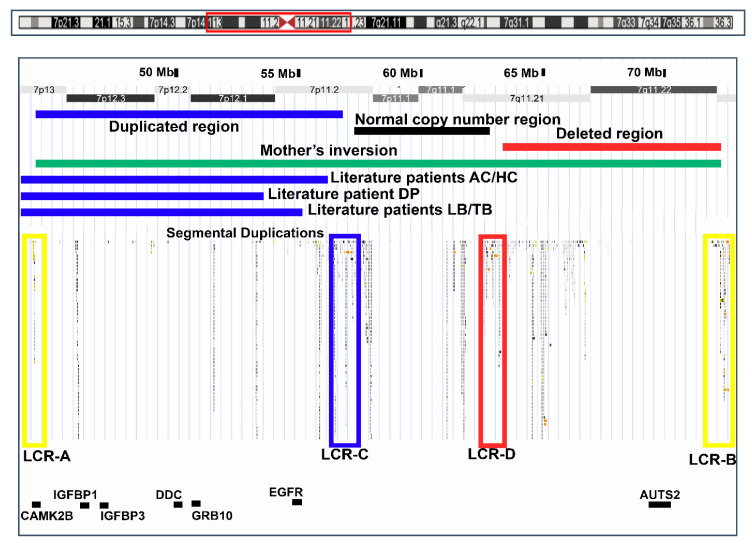
Ideogram of chromosome 7 (top) with the 7p14.1-q11.23 region framed by a red rectangle: and map of this region (UCSC Genome Browser, hg19) (below). Regions involved in the rearrangements in the present case, her mother, and literature cases are shown. Blue bars indicate duplications in the proband and literature cases. The black bar indicates the proband’s normal copy number region. The red bar indicates deletions in the proband. The green bar indicates the inversion in her mother. High homology Low Copy Repeat (LCR) blocks are highlighted as colored empty rectangles. Relevant genes discussed in the present work are presented in black characters.

**Figure 3 ijms-21-08487-f003:**
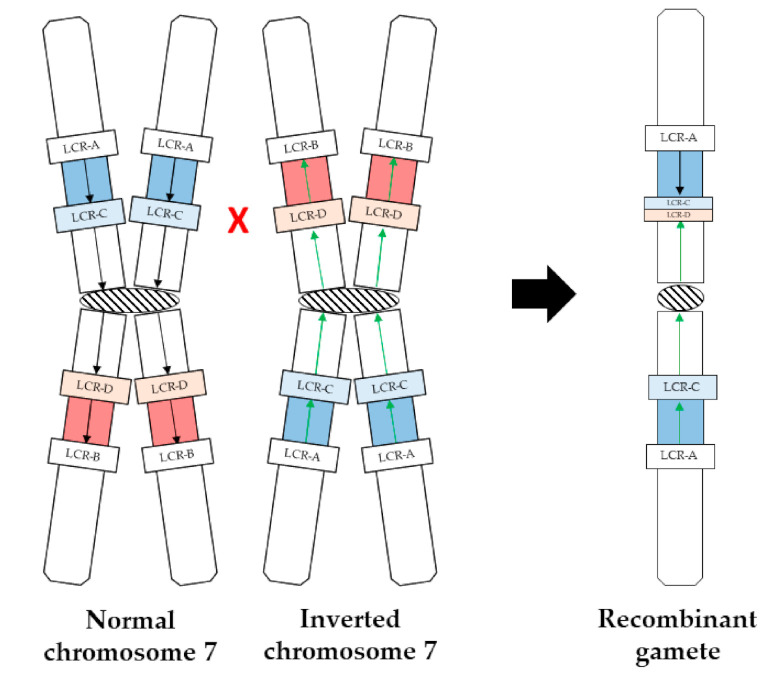
Schematic of the model proposed to explain the mechanism leading to the recombinant gamete. Left: normal and inverted maternal chromosome 7; right: proband’s recombinant chromatid. The black and green arrows indicate, respectively, the normal and inverted orientations of the region involved in maternal inversion. Localization of LCR-A,-B,-C,-D is indicated. The red cross points to the approximate non-allelic homologous recombination (NAHR) site.

**Table 1 ijms-21-08487-t001:** Segregation pattern of informative Single Nucleotide Polymorphisms (SNPs) from the parents to the proband mapped within the proband’s deleted (7q) region. In this region, only SNPs from the father (highlighted by a light blue background) are present.

Region on chr.7	SNP Probes			Genotype						
	SNP_ID	Probe Name	Cytoband	Father		Proband			Mother	
Deleted region	rs6945241	A_20_P00145608	7q11.22	C	T	C			T	T
	rs2103132	A_20_P00247516	7q11.22	C	G	C			G	G
	rs6460543	A_20_P00145611	7q11.22	A	G	A			G	G
	rs7793970	A_20_P00247563	7q11.22	A	G	A			G	G
	rs6979389	A_20_P00145689	7q11.22	C	T	C			T	T

**Table 2 ijms-21-08487-t002:** Segregation pattern of informative microsatellite markers in the duplicated 7p13-p11.2 region. Paternal alleles are presented with a light blue background; maternal alleles are presented with a pink background.

Region on chr.7	Microsatellites				Number of Repeats						
	Marker	Locus	Cytoband	Range	Father		Proband			Mother	
Duplicated region	12	D7S519	7p13	257–285	269	263	269	261	257	257	261
	21	D7S2422	7p12.1	195–227	208	194	208	192	211	211	192
	22	D7S2467	7p12.1	240–248	239	239	239	241	241	241	241
	33	D7S506	7p12.1	117–146	128	112	128	128	128	128	128
	24	D7S2552	7p11.2	232–282	270	256	270	270	276	276	270

**Table 3 ijms-21-08487-t003:** Literature review of 7p maternal duplications encompassing the *GRB10* gene compared with the present case. SRS: Silver-Russell syndrome; 1, SGA: small for gestational age; 2, postnatal growth failure; 3, relative macrocephaly at birth; 4, protruding forehead; 5, body asymmetry; 6, feeding difficulties and/or low BMI (Body Mass Index); PSS, proportionate short stature; DD, developmental delay; ID, Intellectual Disability; LD, learning disability; +, present; −, absent; pat, paternal; mat, maternal; §, mosaic duplication 7p; #, paternal uniparental disomy (UPD) for chromosome 7 in cells with a normal karyotype; y, years; m, months.

Patient	Sex	Age	SRS Diagnostic Criteria						PSS (SDS)	Additional SRS Features	DDs	Duplication 7p				Other CNV (hg19)	References
			1	2	3	4	5	6				Breakpoints on chr.7 (hg19)	Size (Mb)	Inheritance	N. of RefSeq Genes		
Present case	F	17 y 6 m	−	+	+	−	−	−	+ (−2.81)	micrognatia, hypoglycaemia	motor and speech delay, severe ID, head stereotypy	44,114,508-56,786,860	12.67	de novo, mat origin	64	7q11.21q11.23(63,374,309-72,365,957)x1	Present study
AC §	M	4 y	+	+	Na	−	−	na	+ (−2.00)	5th finger clinodactyly	not reported	39,747,723-56,160,689	16.4	mat	83	−	Monk, 2002 [3]
HC § # (AC’s mother)	F	48 y	na	+	Na	−	−	na	+ (−2.00)	5th finger clinodactyly	not reported	Idem	idem	na	idem	−	
DP	F	5 y	−	−	Na	+	−	+	+ (−2.90)	triangular face, micrognatia, down-turned mouth, 5th finger clinodactyly, hypoglycaemia, excessive sweating	mild DD	39,668,287-53,521,622	13.85	de novo, mat origin	69	7p12.1p11.2(52,885,014-54,748,619)x1	Monk 2000; 2002 [2,3]
LB	F	6.3 y	−	na	−	−	−	na	+ (−3.56)	micrognatia, 5th finger clinodactyly	LD	42,000,548-55,129,179	13.12	mat	65	−	Joyce, 1999; Monk, 2002 [3,4]
TB (LB’s mother)	F	adult	na	na	Na	−	−	−	+ (−2.00)	micrognatia, 5th finger clinodactyly	LD	Idem	idem	de novo, pat origin	idem	−	

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
