# Peer review of "Recombinant Chromosome 7 Driven by Maternal Chromosome 7 Pericentric Inversion in a Girl with Features of Silver-Russell Syndrome"

_ijms, 2020, doi:10.3390/ijms21228487_

Round 1

Reviewer 1 Report

Catusi et al reported a girl with recombinant chromosome 7 driven by maternal chromosome 7 pericentric inversion and described about the correlation between her clinical features and Silver-Russell like phenotype and the rearrangement mechanism. The reviewer thinks that this is an important paper for understanding SRS clinical features and the complex rearrangement mechanism on chromosome 7. However, the reviewer has some comments.

Comment 1

Previous reports described that that maternal 7p duplication including GRB10 results in growth delay and SRS features, and the authors evaluated the clinical feature of this patient and previous reported patients. All patients did not satisfy the Netchine-Harbison clinical scoring system and they have only one or two NH-CSS items. Based on this paper, the reviewer cannot think that maternal 7p duplication including GRB10 result in SRS-like phenotype. The maternal duplication detected in this patient and previously reported patients involved many genes other than GRB10. Furthermore, human GRB10 gene codes for various transcripts biallelically expression almost in all tissues. There are possibitilites that abonormal gene expression of genes other than GRB10 laed to the clinical features of this patient. Thus, the reviewer think that the authors should not focus on SRS-like phenotype in this manuscript.

Comment 2

The authors used the words “SRS-like” and “slight PSS”. These words are not objective. The authors should not use these words.

Comment 3

The locus of genes involved duplicated region and deleted region is importtant. The locus of genes described in this text such as GRB10, AUTS2, CAMK2 and DDC  need to be depicted in Figure 2. Furthermore, Figure 2 is indistinct due to small characters. The authors need to modify Figure 2.

Author Response

We thank the reviewer for appreciating the clinical and pathomechanism aspects of the manuscript

Comment 1

Previous reports described that that maternal 7p duplication including GRB10 results in growth delay and SRS features, and the authors evaluated the clinical feature of this patient and previous reported patients. All patients did not satisfy the Netchine-Harbison clinical scoring system and they have only one or two NH-CSS items. Based on this paper, the reviewer cannot think that maternal 7p duplication including GRB10 result in SRS-like phenotype. The maternal duplication detected in this patient and previously reported patients involved many genes other than GRB10. Furthermore, human GRB10 gene codes for various transcripts biallelically expression almost in all tissues. There are possibitilites that abonormal gene expression of genes other than GRB10 laed to the clinical features of this patient. Thus, the reviewer think that the authors should not focus on SRS-like phenotype in this manuscript.

Answer

We agree that the small number of NH-CSS criteria fulfilled by our patient (and by those in the literature) prevents to classify this series of patients as ‘SRS-like’: we removed the SRS-like designation in the title and throughout the manuscript. Accordingly, we simplified the relative paragraph (see lines 140-141) and revised Table 3

Comment 2

The authors used the words “SRS-like” and “slight PSS”. These words are not objective. The authors should not use these words.

Answer

We thank the reviewer for proper criticism. As mentioned above, we removed the terms ‘SRS-like’ and “slight-PSS” from over the text. In the revised Table 3 the entry “PSS” (column 10 ), maintained to indicate the presence of proportional short stature, is objectified in the entity by specifying within brackets the height in – SDS for  each  patient. Summarizing the following lines changed: 4 (title), 15, 16, 27, 43, 152, 305-306 and Table 3 has been modified

Comment 3

The locus of genes involved duplicated region and deleted region is importtant. The locus of genes described in this text such as GRB10, AUTS2, CAMK2 and DDC need to be depicted in Figure 2. Furthermore, Figure 2 is indistinct due to small characters. The authors need to modify Figure 2.

Answer

We agree on the relevance of the genes involved in the duplicated and in the deleted region. Hence by following the reviewer suggestion we have modified Figure 2 in order to have a better resolution and a clear reading of the candidate genes CAMK2B, IGFBP1, IGFBP3, DDC, GRB10, EGFR and AUTS2 for the clinical signs under discussion.

Reviewer 2 Report

The authors present a very interesting case of a patient with a complex chr 7 rearrangement including a duplication of GRB10. The role of GRB10 in growth and development is currently not fully understood, especially because of it`s complex expression and imprinting pattern and  because they are very rare.

Therfore it is of importance to bring the knowledge about these cases to the scientific community.

This might help to ellucidate the role of GRB10 in the future i.e. by narrowing down a critical region.

General: Some parts of the manuscript could benefit from a review by a native speaker.

Additionally, the figures could be in a higher resultion since some words are not readable.

Some minor points.

L32 SRS is associated with all of these symtoms but diagnosed with 4 (NH-CSS).

L36 - I am not aware of any segmental upd(7p)mat

L70 dosage of 0.6 mg/die --> day?

L83 - please indicate genome build (hg19/hg38) and use staard format (chr7:44,224,508-56,786,860 (hg19))

L85 - BAC stands for Bacterial artificial chromosome

L122 Table 2 - im am not sure if SNPs in 7p12.3 can really be regarded as informative

L154 The sentence "when available the head circumference is rproportional to the height" somehow contradicts the observarion of "relative macrocephaly. To which patients is the sentence referring.

L155 In the overlapping regions from Fig 2 it comes obvious, that IGFBP1 and IGFBP3 are includeded in all duplications in the patients reported here. Are the authors aware of any isolated duplcation that includes these genes and not GRB10?

Author Response

L32 SRS is associated with all of these symtoms but diagnosed with 4 (NH-CSS) (lines 32-33).

The sentence has been changed

L36 - I am not aware of any segmental upd(7p)mat

The sentence has been changed

L70 dosage of 0.6 mg/die --> day?

Corrected “die” with “day” (line 72)

L83 - please indicate genome build (hg19/hg38) and use staard format (chr7:44,224,508-56,786,860 (hg19)) (lines 84-85)

We thank the reviewer for his/her correct note: Standard ISCN formula (including genome build) has been used to report CNVs

 L85 - BAC stands for Bacterial artificial chromosome

Corrected “clone” with “chromosome” (line 87) 

L122 Table 2 - im am not sure if SNPs in 7p12.3 can really be regarded as informative

We agree with the reviewer note and modified Table 1 by removing non informative SNPs in 7p12.3. Indeed, Table 1 is designed to show the maternal deletion.

L154 The sentence "when available the head circumference is rproportional to the height" somehow contradicts the observarion of "relative macrocephaly. To which patients is the sentence referring.

As noted by the reviewer the sentence is confusing and has been removed.

 L155 In the overlapping regions from Fig 2 it comes obvious, that IGFBP1 and IGFBP3 are includeded in all duplications in the patients reported here. Are the authors aware of any isolated duplcation that includes these genes and not GRB10?

We specify that no isolated IGFB1, IGFBP3 and EGFR duplications have been described in literature or in DECIPHER database (lines 154-156; DECIPHER database electronic address added, number 6)

Moreover:

  • Figures have been modified by improving the definition of small characters
  • English language has been edited by the MDPI service.

Round 2

Reviewer 1 Report

The authors responsed for my comments and modified the texr approriately. I think that this manuscript sould be accepted.